# Composable Effects for Flexible and Accelerated Probabilistic Programming in NumPyro

**Du Phan** *
Uber AI
phandu@postech.ac.kr

**Neeraj Pradhan** *
Uber AI
npradhan@uber.com

**Martin Jankowiak**
Uber AI
jankowiak@uber.com

## Abstract

NumPyro is a lightweight library that provides an alternate NumPy backend to the Pyro probabilistic programming language with the same modeling interface, language primitives and effect handling abstractions. Effect handlers allow Pyro's modeling API to be extended to NumPyro despite its being built atop a fundamentally different JAX-based functional backend. In this work, we demonstrate the power of composing Pyro's effect handlers with the program transformations that enable hardware acceleration, automatic differentiation, and vectorization in JAX. In particular, NumPyro provides an iterative formulation of the No-U-Turn Sampler (NUTS) that can be end-to-end JIT compiled, yielding an implementation that is much faster than existing alternatives in both the small and large dataset regimes.

## 1 Introduction

Many probabilistic programming languages (PPLs) are embedded as a DSL within a host language. The advantage of embedding is availability of host language infrastructure along with frameworks for automatic differentiation and hardware acceleration. Within the Python community, some examples of embedded PPLs include Pyro [1] and ProbTorch [2] based on PyTorch [3], Edward2 [4] based on TensorFlow, and PyMC3 [5] based on Theano. NumPyro is a package for probabilistic programming built atop JAX [6, 7], which is a high-level tracing library for program transformations (e.g. automatic differentiation, vectorization and JIT compilation) of Python and NumPy functions. Thus NumPyro enables users to write probabilistic programs using familiar NumPy arrays and operations.

NumPyro is built around the same effect handling abstraction as Pyro. Effect handlers provide a way to inject effectful computation into primitive statements in a probabilistic program, e.g. recording the random choices made in an execution trace. In NumPyro these effects can be easily composed with the JAX tracer that operates at the level of NumPy operations and its own set of primitives for control flow. This allows us to expose a modeling language that is the same as in Pyro. Under the hood, inference subroutines can use effect handlers to inspect and modify program behavior and freely compose with JAX transformations to speed up critical subroutines via parallelization and JIT compilation. As an example (see Sec. 3.1), we implement an iterative version of the No-U-Turn Sampler that can leverage JAX's `jit` transformation for end-to-end compilation and optimization by the XLA compiler [8].

## 2 Support for Pyro's Modeling Interface

NumPyro retains the same language primitives and modeling and inference interface as in Pyro. In particular, NumPyro supports `sample` and `param` statements that allow users to designate random

---

* Equal contribution

Preprint. Under review.

| | seed | trace | condition |
|---|---|---|---|
| Primitives Affected | `sample` | `sample`, `param` | `sample` |
| Description | Seeds `fn` with a `PRNGKey`. Every call to `sample` inside `fn` results in splitting of `PRNGKey` to generate a fresh seed for subsequent calls. | Records the input, output, and function calls inside of `sample`, `param` statements in `fn`. | Conditions unobserved `sample` sites in `fn` to values in `param`. |
| Usage | `seed(fn, rng)(...)` | `trace(fn).get_trace(...)` | `condition(fn, param)(...)` |

Table 1: Examples of effect handlers: the primitive statements affected by each handler, the effect added, and usage w.r.t. the original function `fn` with its arguments denoted by ellipsis.

variables and learnable parameters, respectively. It also has effect handlers like `trace`, `replay` and `condition` to provide nonstandard interpretations to these statements. Table 1 lists some commonly used effect handlers.

Effect handlers have emerged as a composable abstraction for program transformations in PPLs [1, 9, 10]. We have found the effect handling abstraction to be particularly useful in designing a common interface to probabilistic programming, despite fundamental differences in the underlying backend. While PyTorch tries to accommodate much of Python's dynamism and facilities for object-oriented programming with mutable objects (e.g. PyTorch optimizers update parameters in-place), JAX encourages a functional style of programming as required by the tracer. As an example, unlike PyTorch, JAX uses a functional pseudo-random number generator [11], which mandates passing an explicit random number generator key (`PRNGKey`) to distribution samplers. In practice, NumPyro inference algorithms take a single `PRNGKey` from the user that is split to generate new keys when passed to downstream functions. This does not result, however, in any change to probabilistic programs formulated in NumPyro, as this splitting mechanism is abstracted into a `seed` handler that operates on `sample` statements (see Table 1).

## 3 Leveraging JAX Transformations in Inference Subroutines

JAX [7] is a Python library that provides a high-level tracer for implementing transformations of programs in Python and NumPy. Currently, three main transformations are available: i) automatic differentiation (`grad`); ii) JIT compilation (`jit`) to multiple backends using XLA; and iii) automatic vectorization (`vmap`). Frostig et al. [7] note that ML workloads are composed of many pure-and-statically-composed (PSC) subroutines that are good candidates for acceleration. This is also true of the inference subroutines that lie at the core of NumPyro. While NumPyro's frontend—i.e. its modeling and inference API—are close to Pyro, we have taken care to ensure that the core inference algorithms and utilities are purely functional so that we can make extensive use of JAX transformations like `jit` and `vmap`. This allows us to implement highly optimized and parallelizable subroutines. We provide two such examples: i) composing `jit` and `grad` to implement an iterative version of the NUTS sampler that can be end-to-end JIT compiled by JAX; and ii) composing `vmap` with effect handlers like `trace` or `condition` to implement vectorized subroutines.

### 3.1 Iterative No-U-Turn Sampler (NUTS)

The No-U-Turn Sampler (NUTS) [12] is an extension of the Hamiltonian Monte Carlo (HMC) algorithm [13, 14], which can efficiently sample from high-dimensional continuous probability distributions. NUTS adaptively sets the trajectory length parameter in HMC, which along with the adaptation of the step size and mass matrix parameters, ensures that HMC runs efficiently on a variety of models without extensive hand-tuning. This provides a highly attractive black-box inference algorithm for PPLs to have in their toolkit.

To JIT compile NUTS sampling, we need to `jit` transform a key component of the algorithm, namely the `BuildTree` subroutine that recursively builds an implicit balanced binary tree by running the `LeapFrog` integrator (Appendix A). While this can be written as a PSC subroutine, tracing it is hard for two reasons. First, the form of the `LeapFrog` integrator requires us to JIT through a gradient

computation. JAX can handle this, since transformations like `jit` and `grad` are composable.[2] Second—and more problematically—the complex control flow of the recursive formulation cannot be traced for JIT compilation in JAX.[3]

An alternative would be to JIT compile a single `LeapFrog` step or the potential energy function.[4] However, drawing a single sample involves many `LeapFrog` steps, and the overhead in terms of Python function dispatch calls is significant. In addition, this approach significantly reduces opportunities for operator fusion in XLA compilation. To get around these obstacles, we propose an *iterative* version of the NUTS algorithm that can be fully JIT compiled. In particular, this involves converting the `BuildTree` procedure into an iterative procedure, paving the way for a NUTS implementation that can take full advantage of XLA acceleration. As we demonstrate in benchmarking experiments in Sec. 4, the result is an algorithm that is much faster than existing implementations. More details on the algorithm are available in Appendix A.

### 3.2 Vectorizing Subroutines With `vmap`

Many utilities and subroutines for inference, e.g. model prediction, Monte Carlo estimation, or running MCMC chains, can be batched to make use of SIMD vectorization. In many frameworks this kind of batching requires laborious manual threading and/or significant cognitive overhead in managing explicit batch dimensions. JAX provides a vectorizing map (`vmap`) transformation that makes it easy to represent batched computations as mapping over function arguments along an outermost axis. This requires no changes to the underlying code but maintains the efficiency of manual batching.

Since JAX transformations are fully composable with Pyro's effect handlers like `seed`, `trace`, and `condition`, and since the latter are implemented within the Python runtime and thus traceable, `vmap` becomes very powerful. As an example, Fig. 1 shows how we can use `vmap` to batch three common computations: i) sampling from the prior; ii) sampling from the posterior predictive distribution; iii) and computing log-likelihoods. Note that without `vmap` we would need to explicitly handle an additional batch dimension within the `logistic_regression` model and the utility functions in Fig. 1b, which is particularly cumbersome for more involved models. As a final example, in Stochastic Variational Inference (SVI), we optimize a loss function that is a Monte Carlo estimate of the Evidence Lower Bound (ELBO). This requires running the model as well as the inference network multiple times, all of which can be elegantly parallelized using `vmap` (see Appendix D).

## 4 Experiments

We compare the performance of NumPyro's NUTS implementation with that of other frameworks (Stan, Edward2, and Pyro) in both the small and large data regimes. Recall that NumPyro's NUTS implementation is end-to-end JIT compiled, while in both Edward2 and Pyro only the potential energy computation is compiled. We use two benchmark models: i) a Hidden Markov Model (HMM) on a small synthetic dataset; and ii) logistic regression on the `Forest CoverType` dataset [15]. Refer to Appendix C for details on the benchmarking experiments.

**Hidden Markov Model** Since we use a small dataset for this experiment, we expect poor performance on the GPU; consequently we limit ourselves to a CPU-only comparison. Note that, although the dataset is small, the potential energy computation involves a loop that can be expensive to differentiate through. From Table 2, we see that for the HMM, NumPyro is around 500X faster than Pyro and 6X faster than Stan. The iterative procedure in Algorithm 2 introduces insignificant overhead, and the end-to-end compilation allows XLA to output highly optimized code.

**Logistic Regression** For this dataset, which contains more than half a million datapoints, GPU acceleration significantly outperforms the CPU, as expected. The time spent in computing gradients in the `LeapFrog` integrator exceeds the time spent building the tree or computing the terminating condition. Since the bottleneck primarily lies in large tensor operations, we expect the difference

---

[2]Note, however, that for example PyTorch's tracing JIT does not allow for this.

[3]This obstacle was also noted in Tran et al. [4]; as a consequence the NUTS implementation in Edward2 uses TensorFlow eager mode.

[4]This is the approach adopted by the NUTS implementation in Pyro.

```
1  def logistic_regression(x, y=None):
2    ndims = np.shape(x)[-1]
3    m = sample('m', Normal(0., np.ones(ndims)))
4    b = sample('b', Normal(0., 1.))
5    return sample('y', Bernoulli(logits=x @ m +
         b), obs=y)
```

(a) A model for logistic regression in NumPyro

```
1  def predict_fn(rng, param, *args):
2    conditioned_model = condition(logistic_regression, param)
3    return seed(conditioned_model, rng)(*args)
4
5  def ll_fn(rng, param, *args):
6    f_traced = trace(predict_fn)
7    obs = f_traced.get_trace(rng, param, *args)['y']
8    return np.sum(obs_node['fn'].log_prob(obs['value']))
```

(b) Utilities for prediction and log-likelihood computation using `seed`, `trace` and `condition` handlers

```
1  # x -> input dataset
2  # samples -> a dict of samples from the posterior distribution
3  # rngs.. -> batch of PRNGKeys
4
5  prior_predictive = vmap(lambda rng: seed(logistic_regression, rng)(x))(rngs_sim)
6  posterior_predictive = vmap(lambda rng, param: predict_fn(rng, param, x))(rngs_pred, samples)
7  log_likelihood = vmap(lambda rng, param: ll_fn(rng, param, x, y))(rngs_pred, samples)
8  exp_log_likelihood = logsumexp(log_likelihood) - np.log(num_samples)
```

(c) Vectorized sampling from the prior and posterior predictive with log-likelihood computation using `vmap`. A complete code listing (including inference) is provided in Appendix B.

Figure 1: A simple logistic regression model. The modeling language is the same as in Pyro.

| Framework | HMM | COVTYPE |
|---|---|---|
| Stan (CPU) | 0.56 | 146.41 |
| Edward2 (CPU) | - | 56.93 |
| Edward2 (GPU) | - | 9.59 |
| Pyro (CPU) | 46.88 | 38.85 |
| Pyro (GPU) | - | 4.36 |
| NumPyro (CPU) | **0.09** | 30.11 |
| NumPyro (GPU) | - | **2.46** |

Table 2: Time per leapfrog step in different frameworks (in milliseconds).

between the various GPU implementations to narrow. Nevertheless, on this problem NumPyro is about 2X faster than Pyro and 4X faster than Edward2 on the GPU.

## 5   Summary

We describe NumPyro, a package for probabilistic programming using Python and NumPy that uses JAX transformations under the hood for hardware acceleration, automatic differentiation, and vectorization. NumPyro has a functional core, where inference subroutines are pure-and-statically composed functions that can be traced by JAX for parallelization and JIT compilation. These subroutines also make use of effect handlers to inspect and transform probabilistic programs. Effect handlers operate on core language primitives within the Python runtime, are transparent to the JAX tracer, and are, therefore, fully composable with JAX's transformations. This composability allows us to offer the same modeling language as Pyro, and at the same time leverage JAX tranformations to parallelize and JIT compile inference subroutines for significant speed ups. In particular we show that the judicious application of these program transformations allows us to implement an iterative version of the NUTS algorithm that offers strong performance on both the CPU (for small models) and the GPU (for larger models).

## 6   Acknowledgments

We would like to thank Noah Goodman for feedback, and the JAX development team, in particular Matthew Johnson and Peter Hawkins, for their invaluable help with many JAX issues and feature requests.

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

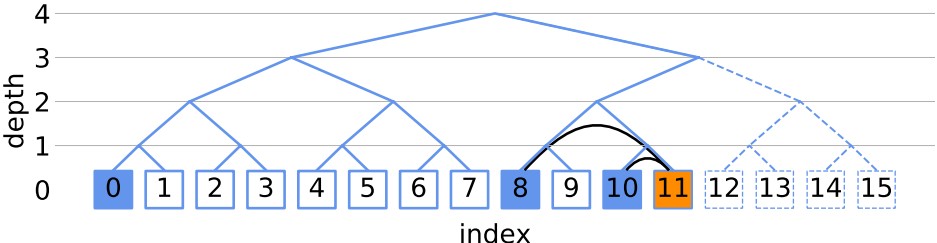

Figure 2: A graphical representation of how binary trees are constructed in ITERATIVEBUILDTREE. The orange node is the leaf generated at the current step. Blue nodes are the leaves stored in memory for the purpose of checking the U-Turn condition. White nodes are past leaves that have been removed from memory. Dashed white nodes have not been generated yet. Thick black lines link the left and right leaves of subtrees where we need to check the U-Turn condition.

## A   Iterative NUTS - Algorithm Details

Computing a trajectory in NUTS involves a doubling procedure where at each iteration we run the `LeapFrog` integrator for twice the number of steps taken in the previous iteration, with the direction (forward or reverse) chosen randomly. This has the effect of building an implicit balanced binary tree. The doubling process is terminated when a subtrajectory from the leftmost to the rightmost node of any balanced subtree begins to double back on itself.

Existing NUTS implementations use a recursive tree building formulation, the `BuildTree` subroutine, to double the trajectory length (see Hoffman and Gelman [12, Algorithm 6]). A simplified version of this subroutine is presented in Algorithm 1. To build a tree at depth $d$, it builds two subtrees at depth $d-1$ and combines them. This recursive procedure also takes care to ensure that memory usage scales as $\mathcal{O}(\log N)$ (where $N = 2^d$) rather than $\mathcal{O}(N)$ by only storing $\mathcal{O}(1)$ data per subtree. This is important because storing all $N$ momentum-position pairs might be prohibitive for large models. The key to JIT compiling NUTS sampling is the ability to convert the recursive `BuildTree` subroutine into an iterative implementation.

The iterative version of `BuildTree` takes an initial node (position-momentum pair) $z_{-1}$ and a tree depth $d$ argument, and runs the `LeapFrog` integrator for $N = 2^d$ steps. We will be using 0-based indexing in the following discussion.

**Checking the U-Turn Condition**   For node $z_n$, we need to check the U-Turn condition with respect to the leftmost nodes of any binary subtree for which $z_n$ is the rightmost node. Let us denote the indices for these candidate nodes by $\mathcal{C}(n)$, and the binary representation of $n$ by $b(n)$. Indices in $\mathcal{C}(n)$ have the same binary representation as $b(n)$ except that trailing contiguous 1s in $b(n)$ are progressively masked by 0; e.g. for $n = 11$, $b(11) = (1011)_2$, and the set of candidate nodes for checking the U-Turn condition are indexed by $\mathcal{C}(11) = \{(1010)_2, (1000)_2\} = \{8, 10\}$. Note that this implies that we only need to check the U-Turn condition at odd-numbered nodes against a subset of previous even-numbered nodes.

This allows us to iteratively build the binary tree by running the `LeapFrog` integrator for $2^d$ steps, and terminating early if the U-Turn condition stated above is satisfied. However, in a naive implementation we would still need $O(N)$ memory, since we would need to store the position-momentum pairs at each step of the integrator, which would be an unacceptable regression from the $\mathcal{O}(\log N)$ memory requirement of the recursive algorithm.

**Memory Efficiency**   To tackle the issue of memory efficiency, we will use an array $S$ to store only even numbered nodes $z_k$ at index $i = \mathtt{BitCount}(k)$. As we iteratively build the tree, nodes in $S$ may be overwritten so that at step $n$, index $i$ stores the largest even node $z_k$ such that $k < n$ and $\mathtt{BitCount}(k) = i$. Note that the maximum size of the array $S$ is $d$ because the largest bit count of even numbers less than $2^d$ is $d-1$.

At an odd step $n$, the data for the candidate nodes indexed by $\mathcal{C}(n)$ must be present in $S$ because the masking procedure ensures that these candidates are the largest even nodes less than $n$ for their corresponding bit counts. Figure 2 illustrates the iterative procedure at step $n = 11$, and full details of the algorithm are provided in Algorithm 2.

| **Algorithm 1** BUILDTREE | **Algorithm 2** ITERATIVEBUILDTREE |
|---|---|
| **Input** initial node $z$, tree depth $d$ | **Input** initial node $z$, tree depth $d$ |
| **if** $d = 0$ **then** | **Initialize** storage $S[0], S[1], ..., S[d-1]$ |
|    $z' \leftarrow$ LEAPFROG$(z)$ | **for** $n \leftarrow 0$ **to** $2^d - 1$ **do** |
|    **return** TREE$(z', z', False)$ |    $z \leftarrow$ LEAPFROG$(z)$ |
| **else** |    **if** $n$ *is even* **then** |
|    $T_L \leftarrow$ BUILDTREE$(z, d-1)$ |      $i \leftarrow$ BITCOUNT$(n)$ |
|    **if** $T_L.turning$ **then** |      $S[i] \leftarrow z$ |
|      **return** $T_L$ |    **else** |
|    **else** |      // gets the number of candidate nodes |
|      $z \leftarrow T_L.right$ |      $l \leftarrow$ TRAILINGBIT$(n)$ |
|      $T_R \leftarrow$ BUILDTREE$(z, d-1)$ |      $i_{max} \leftarrow$ BITCOUNT$(n-1)$ |
|      $z_L \leftarrow T_L.left$ |      $i_{min} \leftarrow i_{max} - l + 1$ |
|      $z_R \leftarrow T_R.right$ |      **for** $k \leftarrow i_{max}$ **to** $i_{min}$ **do** |
|      **if** $T_R.turning$ **then** |        $turning \leftarrow$ ISUTURN$(S[k], z)$ |
|        $turning \leftarrow True$ |        **if** $turning$ **then** |
|      **else** |          **return** TREE$(S[0], z, True)$ |
|        $turning \leftarrow$ ISUTURN$(z_L, z_R)$ | **return** TREE$(S[0], z, False)$ |
|      **return** TREE$(z_L, z_R, turning)$ | |

Figure 3: Comparing the recursive (left) and iterative (right) versions of the tree building algorithm in NUTS. Note that these are high-level specifications of the full algorithms and ignore details about additional metadata in `Tree` such as the proposal candidate. Other details like the step size and choosing between forward and reverse directions are also omitted, since they are not relevant to the proposed changes.

# B    Code for Vectorized Sampling - Logistic Regression

```python
from jax import random, vmap
import jax.numpy as np
from jax.scipy.special import logsumexp

import numpyro.distributions as dist
from numpyro.distributions import Bernoulli, Normal
from numpyro.handlers import condition, seed, trace
from numpyro.mcmc import MCMC, NUTS
from numpyro.primitives import sample

def logistic_regression(x, y=None):
    ndims = np.shape(x)[-1]
    m = sample('m', Normal(0., np.ones(ndims)))
    b = sample('b', Normal(0., 1.))
    return sample('y', Bernoulli(logits=x @ m + b), obs=y)

def predict_fn(rng, param, *args):
    conditioned_model = condition(logistic_regression, param)
    return seed(conditioned_model, rng)(*args)

def loglik_fn(rng, params, *args):
    tr = trace(predict_fn).get_trace(rng, params, *args)
    obs_node = tr['y']
    return np.sum(obs_node['fn'].log_prob(obs_node['value']))

# Generate random data
true_coefs = np.array([1., 2., 3.])
x = random.normal(random.PRNGKey(0), (100, 3))
dim = 3
y = dist.Bernoulli(logits=x @ true_coefs).sample(random.PRNGKey(3))

# Run inference to generate samples from the posterior
num_warmup, num_samples = 500, 500
kernel = NUTS(model=logistic_regression)
mcmc = MCMC(kernel, num_warmup, num_samples)
mcmc.run(random.PRNGKey(1), x, y=y)
samples = mcmc.get_samples()

# Generate batch of PRNGKeys
rngs_sim = random.split(random.PRNGKey(2), num_samples)
rngs_pred = random.split(random.PRNGKey(3), num_samples)

# Prediction and log likelihood
prior_predictive = vmap(lambda rng: seed(logistic_regression, rng)(x))(rngs_sim)
posterior_predictive = vmap(lambda rng, param: predict_fn(rng, param, x))(rngs_pred, samples)
log_likelihood = vmap(lambda rng, param: loglik_fn(rng, param, x, y))(rngs_pred, samples)
expected_log_likelihood = logsumexp(log_likelihood) - np.log(num_samples)
```

Listing 1: Using vmap to vectorize three common inference subroutines: i) sampling from the prior; ii) sampling from the posterior predictive distribution; and iii) computing log-likelihoods.

# C    Experimental Details

All experiments are conducted on a system using an AMD Ryzen Threadripper 1920X processor and an NVIDIA GeForce RTX 2070 graphics card. Framework versions: Stan 2.19, Edward2 0.0.1 (with TensorFlow 1.14 and TensorFlow Probability 0.7), Pyro 0.4.1 (with PyTorch 1.2), NumPyro 0.2 (with JAX 0.1.44 and jaxlib 0.1.27). Code for both benchmarking models can be found on the NumPyro GitHub repo.[5]

**Hidden Markov Model**    To test performance in the small dataset regime, we use an HMM. Following [16, Section 2.6], we construct a semi-supervised HMM model with 3-dimensional latent states and 10-dimensional observations. Using fixed transition and emission matrices, we sample 600 data points and treat the first 100 latent states as observed. For benchmarking we draw $10^5$ NUTS samples for both Stan and NumPyro. Because Pyro's NUTS implementation is extremely slow on this problem, we conduct 5 runs with different random seeds and 100 samples for each run, and then report the average across 5 runs.

---

[5]`https://github.com/pyro-ppl/numpyro/tree/master/examples`

**Logistic Regression**   We consider a logistic regression model on the `Forest CoverType` dataset, which has $581,012$ datapoints and $54$ features. The prior on the weights is a unit normal distribution. Following [4], we normalize all features and transform the multi-class problem into a binary class problem by merging all the classes except for the most frequent one. For benchmarking we draw $100$ samples on the GPU and $30$ samples on the CPU. In addition, we fix the step size in all frameworks to $0.0015$; this value was obtained by running $150$ warmup adaptation steps in NumPyro.

## D   Vectorized Estimation of the Evidence Lower Bound (ELBO) in SVI

```python
from numpyro import optim
from numpyro.svi import SVI, elbo

# model, guide defined externally for the inference problem

# vectorize elbo estimate over `num_particles`
# args are arguments passed by SVI to `elbo`.
def vectorized_elbo(rng, *args, num_particles=100):
  rng = random.split(rng, num_particles)
  return np.mean(vmap(lambda rng_: elbo(rng_, *args))(rng))

optimizer = optim.Adam(1e-3)
svi = SVI(model, guide, vectorized_elbo, optimizer)
```

Listing 2: Using `vmap` to vectorize the computation of the Evidence Lower Bound (ELBO) in SVI.

