# OpenReview forum: "Composable Effects for Flexible and Accelerated Probabilistic Programming in NumPyro"
_NeurIPS.cc/2019/Workshop/Program_Transformations — Program Transformations @NeurIPS2019 Poster_

### Official Review · AnonReviewer1 · 2019-09-26
**Very cool application of program transformations in prob prog**

**Confidence:** 5
**Rating:** 9

**Review:**

the authors describe numpyro, a probabilistic programming system using JAX as a backend. they motivate the software system well, and describe its internals well enough within the page limit. i would have liked to see a simple example program early (first page if possible) to give me a feel of what it's like to write in numpyro.
overall, a real-world system is described, using interesting program transformation technology, with exciting best-in-class benchmarks. this is a very strong submission, and i'd like to hear the authors give a talk on the topic.

---

### Official Review · AnonReviewer2 · 2019-09-28
**Good showcase of compositionality and program transformations**

**Confidence:** 4
**Rating:** 9

**Review:**

This paper presents a reimplementation of the Pyro probabilistic programming system based on Numpy arrays and JAX, a domain specific tracing JIT compiler. This is achieved by composing Pyro’s effect handlers with JAX transformations such as jit and vmap. The paper is written in a clear and accessible language and is easy to follow in general.

As an example of what can be achieved in this composable JIT compilation scheme, the paper presents a new No-U-Turn Sampler (NUTS) inference engine that is much faster than existing implementations due to JIT compilation. The new system is evaluated in two benchmark models, a hidden Markov model (HMM) on a small synthetic dataset and logistic regression with around 0.5M data points. The results indicate that NumPyro’s NUTS implementation is 500 times faster than Pyro and 6 times faster than Stan for the HMM benchmark. For the logistic regression, NumPyro is 2 times faster than Pyro and 4 times faster than Edward2 on the GPU.

The authors state in several places that this new NumPyro implementation is close to Pyro in inference and modeling API. I am curious about how close it really is. Can a Pyro user use NumPyro as a drop-in replacement with the same modeling API, provided that the tensor operations are moved to Numpy arrays?

I think this paper is a very good fit for this workshop, and I would be interested in hearing more about the implementation details.

---

### Decision · Program_Chairs · 2019-10-01

**Decision:**

Accept (Poster)

**Comment:**

Both reviewers thought this was an excellent contribution and we look forward to seeing it presented at our workshop.